# Validity and Reliability of a Novel Smartphone Tele-Assessment Solution for Quantifying Hip Range of Motion

**DOI:** 10.3390/s22218154

**Published:** 2022-10-25

**Authors:** Charlotte J. Marshall, Doa El-Ansary, Adrian Pranata, Charlotte Ganderton, John O’Donnell, Amir Takla, Phong Tran, Nilmini Wickramasinghe, Oren Tirosh

**Affiliations:** 1School of Health Science, Swinburne University of Technology, Hawthorn, VIC 3122, Australia; 2Department of Surgery, School of Medicine, University of Melbourne, Parkville, VIC 3052, Australia; 3Hip Arthroscopy Australia, Richmond, VIC 3121, Australia; 4Department of Orthopaedic Surgery, Western Health, Footscray Hospital, Footscray, VIC 3011, Australia

**Keywords:** tele-assessment, smartphone, hip range of motion

## Abstract

Background: Tele-health has become a major mode of delivery in patient care, with increasing interest in the use of tele-platforms for remote patient assessment. The use of smartphone technology to measure hip range of motion has been reported previously, with good to excellent validity and reliability. However, these smartphone applications did not provide real-time tele-assessment functionality. We developed a novel smartphone application, the TelePhysio app, which allows the clinician to remotely connect to the patient’s device and measure their hip range of motion in real time. The aim of this study was to investigate the concurrent validity and between-sessions reliability of the TelePhysio app. In addition, the study investigated the concurrent validity, between-sessions, and inter-rater reliability of a second tele-assessment approach using video analysis. Methods: Fifteen participants (nfemales = 6) were assessed in our laboratory (session 1) and at their home (session 2). We assessed maximum voluntary active hip flexion in supine and hip internal and external rotation, in both prone and sitting positions. TelePhysio and video analysis were validated against the laboratory’s 3-dimensional motion capture system in session 1, and evaluated for between-sessions reliability in session 2. Video analysis inter-rater reliability was assessed by comparing the analysis of two raters in session 2. Results: The TelePhysio app demonstrated high concurrent validity against the 3D motion capture system (ICCs 0.63–0.83) for all hip movements in all positions, with the exception of hip internal rotation in prone (ICC = 0.48, *p* = 0.99). The video analysis demonstrated almost perfect concurrent validity against the 3D motion capture system (ICCs 0.85–0.94) for all hip movements in all positions, with the exception of hip internal rotation in prone (ICC = 0.44, *p* = 0.01). The TelePhysio and video analysis demonstrated good between-sessions reliability for hip external rotation and hip flexion, ICC 0.64 and 0.62, respectively. The between-sessions reliability of hip internal and external rotation for both TelePhysio and video analysis was fair (ICCs 0.36–0.63). Inter-rater reliability ICCs for the video analysis were 0.59 for hip flexion and 0.87–0.95 for the hip rotation range. Conclusions: Both tele-assessment approaches, using either a smartphone application or video analysis, demonstrate good to excellent concurrent validity, and moderate to substantial between-sessions reliability in measuring hip rotation and flexion range of motion, but less in internal hip rotation in the prone position. Thus, it is recommended that the seated position be used when assessing hip internal rotation. The use of a smartphone to remotely assess hip range of motion is an appropriate, effective, and low-cost alternative to the face-to-face assessments. This method provides a simple, cost effective, and accessible patient assessment tool with no additional cost. This study validates the use of smartphone technology as a tele-assessment tool for remote hip range of motion assessment.

## 1. Introduction

The COVID-19 pandemic led to an exponential increase in digital health services delivered via information and communication technologies [1]. Tele-rehabilitation via online video communication is an emerging area that is attracting increasing attention as a potential alternative to conventional, face-to-face rehabilitation; it is suggested to be an option for people located remotely to reduce the need for frequent travel [2]. A critical missing element in current tele-rehabilitation solutions is the tele-assessment component which supports an objective remote assessment of functional performance—more specifically, the range of motion (ROM) of joints—integrated with web-based management and planning capabilities [3]. The quantification of hip ROM is a key clinical measurement [4,5] that is performed before and after surgical intervention to the hip [6]. The current standard clinical pathway that requires several face-to-face visits over a period of time is costly, and can impose traveling difficulties for isolated and disadvantaged populations, particularly during pandemic events; this can potentially lead to less equitable clinical outcomes and lower patient satisfaction [7].

To address the paucity of tele-assessment solutions to quantify hip ROM, we designed the TelePhysio platform with unique integration methods of a web-based repository system, coupled with the motion sensor inertial measuring unit (IMU) data captured from a smartphone. The use of a portable smartphone to measure hip ROM is not new [8], but the capability to do this remotely through tele-assessment has not, to our knowledge, been investigated previously. 

Recent technological advances have led to the development and use of accelerometer-based smartphone applications to measure knee [9] and hip [8,10,11] ROM. Comparing goniometer and smartphone platforms in the measurement of hip internal rotation, Miyachi et al. [10] reported substantial to almost perfect agreement intra-rater reliability (ICC 0.668–0.939), and a moderate to strong correlation (r = 0.626–0.915) between the smartphone and goniometer methods, demonstrating good criterion-related validity. Charlton et al. [8] have shown moderate to good validity of hip joint ROM assessment using a 3D motion analysis system for flexion, abduction, adduction, internal and external rotation in supine position, and internal and external rotation in the seated position. Takeda et al. [11] reported excellent intra-rater reliability ICCs of 0.87, 0.72, and 0.72 for hip internal rotation, hip external rotation, and hip flexion, respectively. In all the above studies, testing was conducted by a physiotherapist using passive ROM assessment where the examiner passively moved the joint and determined the angle within the pain-free range. The passive method is a limitation when tele-assessment is needed from the patient’s home, as it cannot incorporate the physical interaction of the clinician.

The aim of this study, therefore, was to evaluate the concurrent validity and between-sessions reliability of our tele-assessment TelePhysio app for the evaluation of active hip ROM, to allow remote tele-assessment functionality. In addition, the study investigated the validity, and between-sessions and inter-rater reliability of video analysis as a second tele-assessment method.

## 2. Materials and Methods

### 2.1. Participants

Fifteen healthy adults (nfemales = 6) were recruited for the study. The participants’ mean age was 28.3 ± 4.4 years (22–39 years), their mean weight was 74.3 ± 14.8 kg (45–95 kg), and their mean height was 1.71 ± 0.11 m (1.49–1.91 m). Their eligibility criteria included: 18–40 years of age, English speaking, no history of hip pain/injury or surgery; no history of lower limb injuries in the past three months; not pregnant and showing a negative result for the Flexion-Adduction-Internal Rotation (FADIR) test. Ethical approval was granted by the university ethics committee (ref: 20215539-8106). All participants provided signed written informed consent prior to testing.

### 2.2. Instrumentation

Hip ROM was measured simultaneously using three systems including: (1) the smartphone TelePhysio app; (2) two-dimensional (2D) video camera recording; and (3) three-dimensional (3D) motion capture system (Qualisys Inc, Goteborg, Sweden). TelePhysio was our original tele-assessment app; using the smartphone motion sensors, it allowed clinicians to measure and assess the real-time hip range of their patients in a remote and objective way. At the beginning of the assessment, the assessor remotely connected to the TelePhysio app, which was installed on the participant’s smartphone and strapped around the thigh or shank (Figure 1). The update frequency of the smartphone raw acceleration was set to maximum, which is usually at least 100 Hz for an iPhone (Apple Inc., Cupertino, CA, USA, https://developer.apple.com/documentation/coremotion/getting_raw_accelerometer_events, accessed on 18 October 2022) and an android phone e.g., Samsung (Suwon-si, Korea) and Huawei (Shenzhen, China) [12,13]. Thus, the TelePhysio app was coded to sample at 100 Hz. The smartphones used in this study included: iPhone 8, iPhone 8 plus, iPhone 12 Pro Max, iPhone SE 2020, iPhone X, iPhone 11, iPhone 13, Huawei P10 Lite, Huawei P10, Google pixel 6, Samsung Galaxy s20fe, Samsung Galaxy A8, and Xiaomi POCO X3 NFC. At the end of the assessment, both the assessor and the participant could log in to the web-based application and review the outcomes. The video recording was performed using the webcam embedded in the participant’s laptop with a resolution of 1920 × 1080 and 30 Hz sampling rate.

### 2.3. Protocol

The participants were assessed on two occasions, with an average of 6.4 days (range 4–10 days) between sessions. The first session was conducted face-to-face in the biomechanics laboratory, and the second session was conducted remotely at the participant’s home. The duration of the laboratory session was 45 min, and the at-home session was for 30 min duration. In both sessions, the participant turned on the TelePhysio app and then strapped the smartphone on the selected limb according to the type of assessment (see Section 2.3.1, Section 2.3.2 and Section 2.3.3 for details on strapping). The assessor then connected remotely to the TelePhysio app from their personal web browser and collected motion data whilst the participant performed the hip range task. Once the specific ROM task was completed, the assessor uploaded the smartphone sensor’s data to the web-based cloud application for later processing and analysis. During the assessment, voice communication between the assessor and participant was conducted via Zoom (Zoom Video Communications, Inc., San Jose, CA, USA). The 2D video recording of the participant’s hip range tasks were captured using the participant’s personal laptop computer webcam, which was positioned opposite to them on a table, and the video recording was performed using the Zoom. During the first testing session in the biomechanics laboratory, the 3D motion capture system was used to capture (100 Hz) the 3D movement of six 12 mm reflective markers. Using double-sided tape suitable for skin, the markers were attached to the participant’s left and right greater trochanter, lateral epicondyle of the femur, tibial tuberosity, distal tibia (3 cm above the talocrural joint), lateral malleolus, and the head of the 5th metatarsal (see Figure 1). Hip ROM tests were performed for both right then left legs in the following order: (i) sitting hip internal/external rotation; (ii) prone hip internal/external rotation; (iii) hip flexion in supine; while smartphone was attached anteriorly, then laterally, to the distal thigh (see Figure 1).

#### 2.3.1. Hip Internal and External Rotation during Sitting

The participant was instructed to attach their smartphone antero-laterally to their proximal tibia below the knee and sit with both hips and knees placed at a 90° angle, with feet off the ground (see Figure 1(A1–A3)). The participant was asked to sit in an upright position, thus creating a line from their shoulder to their hip. The participant completed 3 continuous active trials of hip internal and external rotation movements [14].

#### 2.3.2. Hip Internal and External Rotation in Prone

The participant was instructed to attach their smartphone laterally to their tibia above the lateral malleolus and lay prone with their legs in a neutral position, shoulder width apart (see Figure 1(B1–B3)). With the knee of the testing leg flexed to 90°, the participant was asked to internally rotate and then externally rotate their leg, whilst maintaining contact to the floor/mat with the front of their pelvis [15]. Similarly, in the seated position, the participant completed 3 continuous active trials of hip internal and external rotation movements.

#### 2.3.3. Hip Flexion in Supine

The participant was instructed to attach their smartphone to their thigh (anteriorly 5 cm from the top of the patella, and laterally 5 cm above the knee joint) and lay supine while their pelvis was maintained at a level position and parallel to the bed (see Figure 1(C1,C2,D1,D2)). The participant was then asked to flex their hip as far as possible whilst keeping their pelvis level and parallel to the bed. The participant repeated this for 3 trials [15].

### 2.4. Data Analysis

The outcome measures included the maximum inclination angle of the shank (hip internal/external rotation) or thigh (hip flexion) in degrees (o) measured using the three systems: video analysis, TelePhysio, and 3D motion capture (see Figure 1). the hip internal/external rotation was defined as the inclination of the shank segment with respect to the gravity (vertical) line. The hip flexion angle was defined as the inclination of the thigh segment with respect to the ground (horizontal) line. The inclination angles were calculated from: (1) the video files using the video analysis software Tracker (https://physlets.org/tracker/, accessed on 18 October 2022) and its protractor tool; (2) the acceleration signals from the smartphone captured using TelePhysio; and (3) Visual 3D (version v6 2022.08, C-Motion Inc, Kingston, ON, Canada) software from the processed 3D reflecting markers position and software functions. Figure 1 illustrates the testing positions and the measuring angles using the video analysis. To calculate the gravity inclination angle from the smartphone, raw acceleration data were initially filtered at 3 Hz zero phase Butterworth [16] using the Python SciPy library with butter() and filtfilt() functions (Python Language Reference, Version 2.7), followed by the following formula (1) to calculate the angle:(1)$$\emptyset =\mathrm{atan}\;(\frac{\mathrm{accZ}}{\sqrt{{\mathrm{accX}}^{2}+{\;\mathrm{accY}}^{2}}})$$
where Ø is the inclination angle, accX, accY, accZ are acceleration in the x, y, and z axes. The calculated angle from the accelerometer was collected during the static trial. Figure 2 shows an example of the hip angle output and its calculation using the smartphone technology. To calculate the final range, the neutral baseline angle was measured and was subtracted from the maximum hip angle measured at each trial.

The 3D reflective marker positions were processed with Visual3D using the Butterworth low-pass filter at 4 Hz, followed by the following function (2) to calculate the hip range of motion:(2)$$\emptyset =\mathrm{atan}2\left(x,y\right)$$
where Ø is the inclination angle, x and y are, respectively, the vertical and horizontal position of the marker in space. To determine the accuracy of the smartphone and video analysis measurements, each smartphone and video measurement trial and leg side (left and right) was compared with the gold-standard 3D measurement for a particular hip motion in each participant (x3 trials for each measurement and participant).

### 2.5. Statistical Analysis

The statistical comparison of the smartphone TelePhysio measurements with gold-standard 3D measurements was made using an intraclass correlation coefficient [ICC(2,1)], Bland–Altman fixed bias differences and 95% limits of agreement (LOAs), and standard error of measurement (SEM). The interpretation of each ICC was as follows, according to the original definitions by Landis and Koch [17]: 0.00 to 0.20, slight correlation; 0.21 to 0.40, fair correlation; 0.41 to 0.60, moderate correlation; 0.61 to 0.80, substantial correlation; and 0.81 to 1.00, almost perfect correlation. The SEM was calculated for each measurement modality as an additional measure of absolute reliability or validity. This was calculated as described by Atkinson and Nevill [18], as $\mathrm{SEM}=\mathrm{SD}\;\sqrt{1-\mathrm{ICC}}$ where SD is the standard deviation. The resulting SEM value is expressed in degrees with lower values and indicates better absolute validity/reliability. The interpretation of each ICC for the reliability was as described above. Similar analysis was performed to compare the 2D video analysis with the gold-standard 3D measurements. Prior to analysis, the data was checked for the presence of heteroscedasticity using Kendal rank correlation coefficient τ. When τ > 0.1, a logarithmic transformation was performed on the data before calculating bias and 95% upper and lower limits of agreement. 

The TelePhysio and video between-sessions reliability was assessed using ICC(2,1) and SEM. In addition, the minimal detective change (MDC) was calculated to investigate the measurement error to show the limit range in which the amount of change in the two measured values, obtained by repeated measurements, was due to measurement error. Changes larger than the MDC were judged to be “true changes”. The MDC was calculated using the following formula: $\mathrm{MDC}=1.96\sqrt{2\mathrm{SEM}}$. For all analyses, the means between measurements were compared by use of 1-way analysis of variance (ANOVA), with a Scheffe post hoc test to assess for statistical significance at a 0.05 level. All analyses were performed using the free software Statistical Package R version 4.2.1 (https://www.r-project.org/, accessed on 18 October 2022).

## 3. Results

### 3.1. Concurrent Validity

A comparison of TelePhysio measurements with the 3D measurements, to determine the agreement between the two systems, is presented in Table 1 and Figure 3. All hip motion assessments, with the exception of hip internal rotation at prone, showed substantial agreement between TelePhysio and 3D analysis (ICCs > 0.6). The hip flexion assessments had the highest almost perfect agreement (ICCs > 0.80), resulting in similar ICC values when the smartphone was placed anteriorly or laterally on the thigh (0.81, *p* = 0.12 and 0.83, *p* = 0.38, respectively). The hip internal/external rotation assessments showed lower ICC compared to hip flexion, ranging from 0.48 for internal rotation at prone to 0.81 external rotation at prone. The average values of the mean differences for the hip assessments were 1.70° ± 1.25 (0° to 3.1°). The hip internal and external rotation means in sitting, between the TelePhysio and 3D measurements, were significant: *p* < 0.05. The Bland–Altman fixed bias average was −0.02° (−0.11° to 0.13°), and 95% LOA average range −0.53° to 0.49° (see Figure 3). This shows the tendency of the TelePhysio, in measuring hip range, to be 0.02° less than the 3D analysis measurement.

A comparison of the video analysis measurements with the 3D measurements, to determine the agreement between the two systems, is presented in Table 2 and Figure 3. All hip motion assessments, with the exception of hip internal rotation at prone, showed almost perfect agreement (ICCs > 0.8). Like TelePhysio, the hip internal rotation at prone, when measured with the video, had a fair agreement with the 3D analysis (ICC = 0.44, *p* = 0.01). The average values of the mean differences for the hip assessments were 0.80° ± 1.83 (0.8° to 3.4°). Hip internal rotation in prone and hip flexion means between the video analysis and 3D analysis were significant: *p* < 0.01. The Bland–Altman fixed bias average was 0.73° (3.35° to −0.48°), and 95% LOA average range −11.42° to 18.13° (see Figure 3). This shows the tendency of the video analysis method, in measuring hip range, to be 0.73° less than the 3D analysis.

### 3.2. Between-Sessions Reliability

Table 3 shows the between-sessions reliability of TelePhysio. ICCs for hip flexion were found to have substantial agreement values of 0.63 (95%CI 0.35–0.81) and 0.64 (95%CI 0.37–0.81) for the anterior and lateral attachments, respectively. The internal rotation assessments for both sitting and prone positions showed lower ICCs with fair agreement of 0.47 (95%CI 0.13–0.71) and 0.44 (95%CI 0.09–0.69), respectively, while the external rotation was found to have greater ICCs of 0.63 (95%CI 0.35–0.81) and 0.50 (95%CI 0.15–0.73), respectively. The SEMs ranged from 4.43° to 7.05° with average MDC of 6.37° (5.83° to 7.36°).

Between-sessions reliability of the video analysis is presented in Table 4. ICCs for hip external rotation were 0.54 (95%CI 0.19–0.76) and 0.60 (95%CI 0.31–0.79) for the sitting and prone positions, respectively. The internal rotation assessments for both sitting and prone positions were lower with ICCs indicating fair agreement of 0.36 (95%CI −0.01–0.64) and 0.52 (95%CI 0.19–0.76), respectively. Hip flexion ICC was 0.52 (95%CI −0.01–0.77). The SEMs ranged from 4.08° to 7.18° with average MDC of 6.43° (5.59° to 7.43°).

### 3.3. Inter-Rater Reliability

Table 5 shows the results of inter-rater reliability. The ICCs of the hip internal rotation were 0.93 (*p* = 0.32, 95%CI 0.88–0.96) and 0.95 (*p* = 0.48, 95%CI 0.92–0.97) for sitting and prone positions, indicating almost perfect inter-rater reliability, and that of the hip external rotation was 0.87 (95%CI 0.47–0.95) and 0.94 (95%CI 0.89–0.96), respectively, indicating similar almost perfect inter-assessor reliability. Hip flexion showed lower ICC of 0.59 (*p* > 0.01, 95%CI −0.07–0.86) indicating substantial agreement. For hip rotation, the SEMs ranged from 1.88° to 2.54°, and for hip flexion, the SEM was 6.28°.

## 4. Discussion

The aim of this study was to investigate the concurrent validity and between-sessions reliability of two tele-assessment approaches for measuring hip rotation and flexion ROM. The first approach of tele-assessment involved the use of video analysis to capture the task, using a laptop webcam via Zoom. The second approach involved the use of a novel smartphone application, the TelePhysio app, that allows real-time remote data capture from smartphone accelerometer sensors. The reason for exploring the validity and reliability of both 2D video analysis and TelePhysio was to allow clinicians to understand the advantages and disadvantages of each technique, and make better decisions when applying their tele-assessment protocol.

Both tele-assessment approaches were found to have substantial agreement when compared to the gold-standard of the 3D motion capture system. Keogh et al. [19] reported that ICC > 0.75, LOA < ±9.8° and SEM < 5° meet the criteria of validity. The majority of the smartphone measurements were found to be valid with ICC > 0.63, LOA < ±9.8°, and SEM < 5°. Similar to the video analysis method, the smartphone hip internal rotation in prone was the least valid, having ICC = 0.48, LOA ± 13° and SEM = 4.93°. This suggests that the sitting position, compared to the prone, may be a better choice when assessing the hip internal rotation range.

Overall, the smartphone TelePhysio app showed an average of up to 3° lower values (bias) compared to the gold-standard 3D analysis. The study outcomes are in agreement with previous studies that explored the validity of a smartphone bubble inclinometer app and found ICC values > 0.7 [8,10,11,20]. Ganokroj et al. [20] reported ICCs of 0.89, 0.70, 0.81, and 0.9 for sitting internal, sitting external, prone internal, and prone external hip rotation, with SEM values of 2.9°, 3.5°, 4.3°, and 4.2°, respectively. Charlton et al. [8] reported ICCs in sitting internal and external rotation of 0.84 an 0.63 and SEM values of 3.4° and 2.8°, respectively. The differences between the studies reported by Ganokroj et al. [20] and Charlton et al. [8] and our study may be related to the positioning of the smartphone on the tibia. In our study, the smartphone was attached laterally at the mid tibia, while in other studies the smartphone was attached in a proximal-anterior position.

The between-sessions reliability of both tele-assessment approaches for hip ROM measurements are considered good. All hip ROM measurements, with the exception of hip external rotation in prone, showed a MDC smaller than 6.5°. Hip external rotation in prone showed MDC of 7.43° for video analysis, and 7.36° for TelePhysio. According to Hamersma et al. [21], the %MDC is a better measure when considering the reliability of a measuring tool, with %MDC < 20% to be acceptable. The %MDC value is calculated by dividing the MDC by the measured value. In this study, for hip external rotation in prone, we divided the MDC by the mean value and found %MDC values to be 18% and 20% for 2D video analysis and smartphone, respectively. Therefore, the video analysis and the TelePhysio measurement methods can be considered acceptable.

Compared to Charlton et al. [8], our ICC values for hip internal and external rotation in the sitting position were lower. Charlton et al. [8] reported ICCs of 0.84 and 0.63 for internal and external rotation in sitting, respectively, compared to this study that found video analysis ICCs of 0.36 and 0.54, and TelePhysio ICCs of 0.47 and 0.63 for internal and external rotation, respectively. Similarly, higher ICCs were reported by Ganokroj et al. [20] of 0.97, 0.89, 0.91, and 0.90 for sitting internal, sitting external, prone internal, and prone external rotation, with SEM values of 1.33°, 1.85°, 3.2°, and 2.80°, respectively.

There are several reasons for the lower ICCs found in our study, compared to previous studies. The first may be related to the differences in the testing protocol where, in our study, the hip range was active (i.e., the participant moved the limb without any assistant from the assessor), while in previous studies, the hip range was more controlled with passive movement (i.e., the clinician held the participant’s ankle of the tested leg with one hand, while a second hand fixed the knee or pelvic). The second reason may be related to the differences in the locations between the first and second sessions. Previous studies reported both testing sessions as being in a well-controlled laboratory setting. In our study, the first session was performed in the laboratory, accompanied by the clinician to monitor participant compliance, whilst the second session was performed in a less-controlled environment, remoting into the participant’s residence without face-to-face clinician supervision. Therefore, during the second session, the participant’s dynamic movements were less regulated and thus had the potential to introduce greater error. The aim of our study was to explore the reliability of the methods for the use of tele-assessment sessions to accommodate a more realistic and real-world setting.

The present study found that hip internal rotation in prone is the least valid assessment for both modes of tele-health assessment. Contrary to other studies, which reported good to excellent agreement for the measurement of hip internal rotation [8,20]. We suggest that clinicians adopt the seated position when assessing the hip internal rotation range of motion via tele-health. Overall, TelePhysio provides a simple method to connect a clinician with their patients to remotely measure their range of motion. The advantages include: (1) low cost; (2) convenience; and (3) remote and quick assessment. Moreover, the platform has the potential to allow the patient to perform self-assessment and monitoring without clinician intervention, and to automatically upload the data to the web-based cloud application after each test, such that the clinician can review the testing outcomes.

Our study has a number of limitations. Firstly, participants used a range of personal smartphone devices, and this may introduce potential error due to data sampling from the variation in smartphone devices, operating software platforms, and motion sensors. Secondly, the study population was limited to healthy young adults, and future research needs to widen the range of participants with respect to age and cultural and linguistic diversity. Thirdly, active movements initiated by the participant were used for hip range of motion testing and not the passive range of movement, which can be clinically important. Fourthly, the end points of each measurement can be affected by movement from other segments such as the pelvis or trunk; this is a limitation that cannot be controlled on a tele-platform and needs to be visually inspected during the real-time video conference. Nevertheless, the protocol was found to have MDC < 6°, which is encouraging and acceptable within clinical settings.

## 5. Conclusions

Remote tele-assessment, using a smartphone application, has good to excellent validity, and moderate to substantial inter-session reliability in the measurement of hip rotation and flexion range of motion, but less for internal hip rotation in the prone position. Thus, we recommend assessing hip internal rotation in the seated position. The use of a smartphone to remotely assess hip range of motion is an appropriate, effective, and low-cost alternative to face-to-face assessments. This method provides a simple, cost-effective, and accessible patient assessment tool. This study validates the use of smartphone technology as a tele-assessment tool for remote hip range of motion assessment. The use of a smartphone to remotely assess hip range of motion has the potential to improve remote patient care.

## Figures and Tables

**Figure 1 sensors-22-08154-f001:**
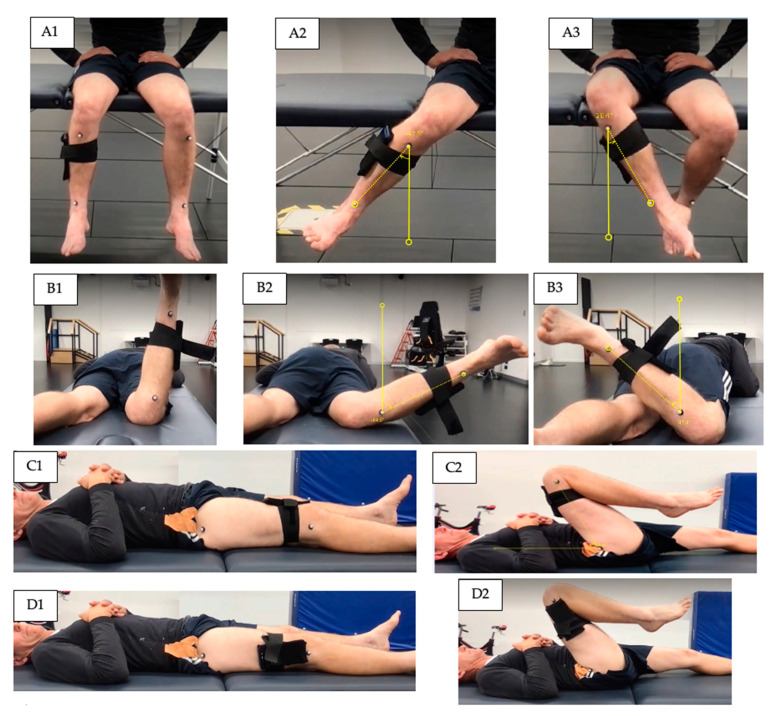
Range of motion measurements using the TelePhysio app, 2D video recording and analysis tool, and 3D motion capture reflective marker system. Starting positions (**A1**,**B1**,**C1**,**D1**). Hip internal rotation sitting (**A2**) and prone (**B2**). Hip external rotation sitting (**A3**) and prone (**B3**). Hip flexion with smartphone anterior (**C2**) and lateral (**D2**) position.

**Figure 2 sensors-22-08154-f002:**
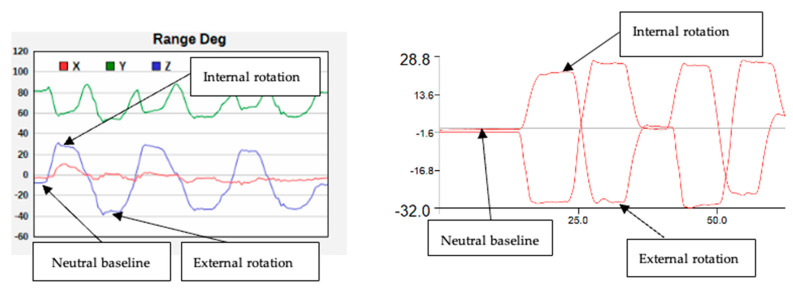
Estimating hip range using the smartphone (**left**) and the 3D motion capture (**right**). To calculate the maximum hip range the neutral baseline angle was subtracted from the maximum internal or external rotation values.

**Figure 3 sensors-22-08154-f003:**
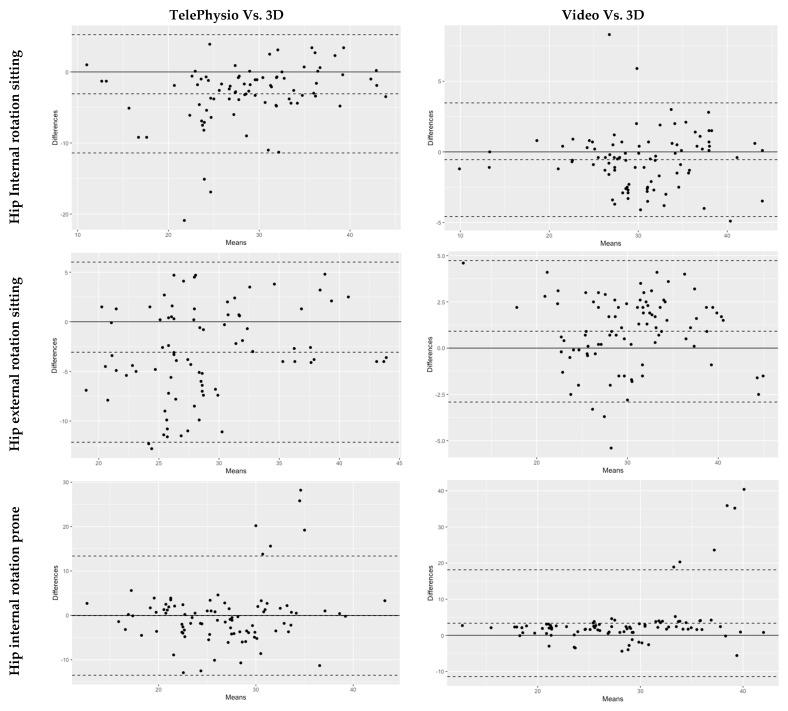
Bland-Altman plots showing agreement between TelePhysio and 3D measurements, and the agreement between Video and 3D measurements for hip range assessments. The y-axis represents the differences (degrees) between the two methods. The x-axis represents the means (degrees) of the two methods. The horizonal solid black line is zero. The horizontal middle dash line is the mean difference (bias) between the methods. The upper horizontal dash line is +1.96 upper limit, and the lower horizontal line is the −1.96 lower limit.

**Table 1 sensors-22-08154-t001:** Comparison of smartphone (TelePhysio) measurements and gold-standard (3D analysis) measurements: means ± standard deviations, one way ANOVA *p*-value, ICC and 95% confidence intervals, Bland–Altman fixed bias and 95% LOA, and SEM.

Measurement	Tele Mean ± SD (°)	3D Mean ± SD (°)	*p*-Value	ICC −95%–+95% CI	Bias ± SD Lower–Upper LOA (°)	SEM (°)
Hip Internal rotation sitting	27.3 ± 7.66	30.4 ± 6.47	0.007	0.75 0.41–0.87	−0.11 ± 0.18 −0.53–0.29	2.12
Hip external rotation sitting	27.4 ± 6.59	30.5 ± 5.49	0.04	0.63 0.30–0.79	0.13± 0.18 −0.21–0.49	2.81
Hip internal rotation prone	26.6 ± 7.28	26.6 ± 6.14	0.99	0.48 0.31–0.63	−0.11 ± 0.16 −0.21–0.44	4.93
Hip external rotation prone	38.2 ± 9.45	38.9 ± 10.4	0.87	0.81 0.73–0.87	−0.01 ± 0.18 −0.28–0.38	2.60
Hip flexion smartphone anterior position	66.8 ± 6.90	68.9 ± 6.13	0.12	0.81 0.62–0.89	−0.03 ± 0.05 −0.13–0.07	1.55
Hip flexion smartphone lateral position	66.8 ± 7.28	68.1 ± 5.84	0.38	0.83 0.73–0.89	−0.02 ± 0.05 −0.12–0.10	1.48

**Table 2 sensors-22-08154-t002:** Comparison of video measurements and gold-standard (3D analysis) measurements: means ± standard deviations, one way ANOVA *p*-value, ICC and 95% confidence intervals, Bland–Altman fixed bias and 95% LOA, and SEM.

Measurement	Video Mean ± SD (°)	3D Mean ± SD (°)	*p*-Value	ICC −95%–+95% CI	Bias Lower–Upper LOA (°)	SEM (°)
Hip Internal rotation sitting	29.9 ± 6.50	30.4 ± 6.47	0.86	0.94 0.91–0.96	−0.48 ± 2.06 −4.53–3.56	0.50
Hip external rotation sitting	31.3 ± 5.69	30.5 ± 5.49	0.61	0.93 0.87–0.95	0.82 ± 1.93 −2.97–4.62	0.51
Hip internal rotation prone	30.0 ± 8.58	26.6 ± 6.14	0.01	0.44 0.23–0.61	3.35 ± 7.54 −11.42–18.13	5.64
Hip external rotation prone	38.4 ± 10.2	38.9 ± 10.4	0.93	0.92 0.89–0.95	−0.02 ± 0.09 −0.17–0.21	1.08
Hip flexion	65.9 ± 6.70	68.5 ± 5.98	0.0007	0.85 0.29–0.94	−0.04 ± 0.04 −0.12–0.04	0.96

**Table 3 sensors-22-08154-t003:** Comparison of smartphone (TelePhysio) measurements in the laboratory and at home: means ± standard deviations, one way ANOVA *p*-value, ICC and 95% confidence intervals, SEM, and MDC.

Measurement	Laboratory Mean ± SD (°)	Home Mean ± SD (°)	*p*-Values	ICC −95%–+95% CI	SEM (°)	MDC (°)
Hip Internal rotation sitting	27.3 ± 7.66	25.8 ± 7.20	0.43	0.47 0.13–0.71	5.37	6.42
Hip external rotation sitting	27.4 ± 6.59	25.5 ± 8.08	0.34	0.63 0.35–0.81	4.43	5.83
Hip internal rotation prone	26.6 ± 7.28	27.9 ± 7.68	0.46	0.44 0.09–0.69	5.54	6.52
Hip external rotation prone	38.2 ± 9.45	37.0 ± 10.7	0.72	0.50 0.15–0.73	7.05	7.36
Hip flexion smartphone anterior position	66.8 ± 6.90	68.4 ± 8.32	0.18	0.63 0.35–0.81	4.74	6.03
Hip flexion smartphone lateral position	66.8 ± 7.28	70.5 ± 8.27	0.94	0.64 0.37–0.81	4.86	6.11

**Table 4 sensors-22-08154-t004:** Comparison of video analysis measurements in the laboratory and at home: means ± standard deviations, one way ANOVA *p*-value, ICC and 95% confidence intervals, SEM, and MDC.

Measurement	Laboratory Mean ± SD (°)	Home Mean ± SD (°)	*p*-Values	ICC −95%–+95% CI	SEM (°)	MDC (°)
Hip Internal rotation sitting	29.9 ± 6.40	30.1 ± 6.22	0.91	0.36 −0.01–0.64	5.00	6.20
Hip external rotation sitting	31.3 ± 5.50	34.2 ± 6.26	0.07	0.54 0.19–0.76	4.08	5.59
Hip internal rotation prone	29.8 ± 8.68	28.9 ± 8.45	0.71	0.52 0.19–0.74	5.37	6.42
Hip external rotation prone	38.6 ± 10.0	40.6 ± 10.80	0.46	0.60 0.31–0.79	7.18	7.43
Hip flexion	65.9 ± 6.55	72.0 ± 8.25	0.0002	0.52 −0.01–0.77	5.60	6.55

**Table 5 sensors-22-08154-t005:** Comparison of 2D video analysis measurements from assessor 1 and assessor 2 during home assessment: means ± standard deviations, one way ANOVA *p*-value, ICC and 95% confidence intervals, and SEM.

Measurement	Assessor 1 Mean ± SD (°)	Assessor 2 Mean ± SD (°)	*p*-Values	ICC −95%–+95% CI	SEM (°)
Hip Internal rotation sitting	29.6 ± 7.35	30.1 ± 6.22	0.32	0.93 0.88–0.96	1.88
Hip external rotation sitting	36.4 ± 6.79	34.2 ± 6.26	0.019	0.87 0.47–0.95	2.40
Hip internal rotation prone	27.7 ± 9.38	28.9 ± 8.45	0.48	0.95 0.92–0.97	2.04
Hip external rotation prone	39.2 ± 9.86	40.6 ± 10.80	0.37	0.94 0.89–0.96	2.54
Hip flexion	62.9 ± 9.17	71.8 ± 8.37	> 0.0001	0.59 −0.07–0.86	6.28

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
