# Peer review of "Validity and Reliability of a Novel Smartphone Tele-Assessment Solution for Quantifying Hip Range of Motion"

_sensors, 2022, doi:10.3390/s22218154_

Round 1

Reviewer 1 Report

line 43: I would add "tele-medicine" and "inertial sensors" between the keywords.

line 53: I would add the following review to support this statement (it is about the use of wearable sensors for remote assessment and monitoring of human movement for clinical applications, and there is a chapter reserved to the use of smartphones): https://doi.org/10.1080/17434440.2021.1988849

line 92: please move the description of the sample (age, mass, stature etc...) from the results section to here.

line 103: since the app relies on the smartphone's sensors, a note on the model(s) of smartphones should be added. What was the sampling freq of the accelerometer? You should also declare the sampling frequency and the resolution of the webcam(s) you used.

line 117. strapped where? Please specify.

line 159: you should define joint angles clearly. For eg., for the accelerometer-based measurement "Hip flexion is defined as the inlincation of the thigh segment with respect to the gravity line (or maybe with respect to the horizontal line....but it does not make any difference) in the lateral (sagittal) plane as measured when lying supine. Hip internal/external rotation is defined as the inclination of the shank with respect to the gravity in the frontal plane when lying prone or supine." Currently you write "inclination angle of the shank (hip internal/external rotation)" but it does not mean anything. Figure no. 2 helps in understanding, but you also need to define these angles in the text. Moreover, I guess that these joint angles are computed slightly differently from a system to another (for eg., hip flexion using fotogrammetry (2D) or stereophotogrammetry (3D) I guess is computed as the inlcination of the thigh with respect to the trunk instead of the gravity/horizontal line). If so, please specify.

line 167: please specify that this "inclination angles" is referred to the gravity line.

line 168: 3 Hz of cut-off comes from where? Any reference? Please add if so.

line 168: "python" is the name of a program. Please refer to it properly in the text, es. "Python (Python Software Foundation. Python Language Reference, version XX.X)"  

line 173: please specify that these accelerometers readings are collected during a static trial.

line 208 (Figure 2 caption): please do not use simply "3D" for representing the Qualisys system (you could use "3D motion capture" or even a  typical used acronym such as "OMC" - optical motion capture).

Generally, I do not understand why comparing against the 2D video system when you have 3D motion capture as a gold standard. Please specify in the discussion. 

statistical analysis: please not that you need to check for the presence of heteroscedasticity in the data (you can use the Kendall rank correlation coefficient τ or visual inspection of the B&A plot). When τ > 0.1 a logarithmic transformation of the data needs to be performed before calculating bias and 95% upper and lower limits of agreement. Basically, if you see a linear trend in the B&A plot you need to perform a log transform. Moreover, did you check normal distribution of data? Please specify.

line 380: any idea why hip internal rotation in prone has the worst accuracy both with acc and 2D video? Please comment.

Author Response

Reviewer 1

  • line 43: I would add "tele-medicine" and "inertial sensors" between the keywords.

Author response: Thank you for that suggestion. We included both keywords in the manuscript. See line 43

  • line 53: I would add the following review to support this statement (it is about the use of wearable sensors for remote assessment and monitoring of human movement for clinical applications, and there is a chapter reserved to the use of smartphones): https://doi.org/10.1080/17434440.2021.1988849

Author response: Thank you for drawing our attention to this interesting article. We included and referenced the article as suggested. See line 53

  • line 92: please move the description of the sample (age, mass, stature etc...) from the results section to here.

Author response: As suggested. We moved the sentence “Participants mean age was 28.3 ± 4.4 years (22-39 years), weight 74.3 ± 14.8 kg (45-95 kg), and height 1.71 ± 0.11 m (1.49-1.91 m).” from the results to line 92.

  • line 103: since the app relies on the smartphone's sensors, a note on the model(s) of smartphones should be added. What was the sampling freq of the accelerometer? You should also declare the sampling frequency and the resolution of the webcam(s) you used.

Author response: Thank you for drawing our attention to this. We included the list the smartphones used by participants and explained the sampling frequency by adding the following sentences: “The update frequency of the smartphone raw acceleration was set to maximum, which is usually at least 100 Hz for iPhone (Apple Inc, https://developer.apple.com/documentation/coremotion/getting_raw_accelerometer_events) and android e.g. Samsung and Huawei [12, 13]. Thus, the TelePhysio app was coded to sample at 100Hz. The smartphones used in this study include: iPhone 8, iPhone 8 plus, iPhone 12 Pro Max, iPhone SE 2020, iPhone X, iPhone 11, iPhone 13, Huawei P10 Lite, Huawei P10, Google pixel 6, Samsung Galaxy s20fe, Samsung Galaxy A8, and Xiaomi POCO X3 NFC. At the end of the assessment, both the assessor and the participant can login to the web-based application and review the outcomes. The video recording was performed using the webcam embedded in the participant’s laptop with a resolution of 1920x1080 and 30 Hz sampling rate.” See line 108

  • line 117. strapped where? Please specify.

Author response: Thank you for that comment. The location of the strap depends on the type of assessment and is later specified in 2.3.1-2.3.3. Thus we added the following  “…. (see 2.3.1 – 2.3.3 for details on strapping)”. See line 126.

  • line 159: you should define joint angles clearly. For eg., for the accelerometer-based measurement "Hip flexion is defined as the inlincation of the thigh segment with respect to the gravity line (or maybe with respect to the horizontal line....but it does not make any difference) in the lateral (sagittal) plane as measured when lying supine. Hip internal/external rotation is defined as the inclination of the shank with respect to the gravity in the frontal plane when lying prone or supine." Currently you write "inclination angle of the shank (hip internal/external rotation)" but it does not mean anything. Figure no. 2 helps in understanding, but you also need to define these angles in the text. Moreover, I guess that these joint angles are computed slightly differently from a system to another (for eg., hip flexion using fotogrammetry (2D) or stereophotogrammetry (3D) I guess is computed as the inlcination of the thigh with respect to the trunk instead of the gravity/horizontal line). If so, please specify.

Author response: Following the reviewer suggestion we clarified the joint angle definition. We added the following sentence “Hip internal/external rotation was defined as the inclination of the shank segment with respect to the gravity (vertical) line. Hip flexion angle was defined as the inclination of the thigh segment with respect to the ground (horizontal) line.” (see line 171). Further explanation to the calculation using the accelerometer data and the 3D position of the reflective markers using the 3D motion capture data, are added in section 2.4

  • line 167: please specify that this "inclination angles" is referred to the gravity line.

Author response: Thank you for that suggestion. The word gravity was added to verify the reference. See line 180

  • line 168: 3 Hz of cut-off comes from where? Any reference? Please add if so.

Author response: We have added the reference to the 3 Hz cut-off. See line 181.

  • line 168: "python" is the name of a program. Please refer to it properly in the text, es. "Python (Python Software Foundation. Python Language Reference, version XX.X)"

Author response: Thank you for drawing our attention to this oversight. We included Python details Python Software Foundation. Python Language Reference, version 2.7)". see line 182

  • line 173: please specify that these accelerometers readings are collected during a static trial.

Author response: We followed the reviewer's suggestion and included the sentence “The calculated angle from the accelerometer are collected during static trial.” See line 188.

  • line 208 (Figure 2 caption): please do not use simply "3D" for representing the Qualisys system (you could use "3D motion capture" or even a typical used acronym such as "OMC" - optical motion capture).

Author response: As suggested we clarified that this is 3D motion capture and added the words “3D motion capture”.

  • Generally, I do not understand why comparing against the 2D video system when you have 3D motion capture as a gold standard. Please specify in the discussion.

Author response: Thank you for that comment. In general, we were interested in measuring hip range by means of tele-assessment. The use of 2D video is one technique that our clinicians currently use, thus we were interested to explore the validity and reliability of such tele-assessment technique. The use of smartphone with the TelePhysio app is the new tele-assessment technique that we were also interested to explore its validation and reliability. Therefore, we thought of publishing the 2D video validity for those that are interested in using video analysis for their tele-assessment. We included the following sentences in the discussion: “The reason for exploring the validity and reliability of both 2D video analysis and TelePhysio was to allow clinicians understand the advantages and disadvantages of each technique and make better decision when applying their tele-assessment protocol.”. See lines 455-458.

  • statistical analysis: please not that you need to check for the presence of heteroscedasticity in the data (you can use the Kendall rank correlation coefficient τ or visual inspection of the B&A plot). When τ > 0.1 a logarithmic transformation of the data needs to be performed before calculating bias and 95% upper and lower limits of agreement. Basically, if you see a linear trend in the B&A plot you need to perform a log transform. Moreover, did you check normal distribution of data? Please specify.

Author response: Thank you for drawing our attention to this oversight. As suggested we re-analysed the data and checked for the presence of heteroscedasticity using Kendal rank correlation coefficient, and when it was > 0.1 we performed log transform of the data and recalculated the bias and 95% lower and upper limits. We added the following sentence: “Prior to analysis the data was checked for the presence of heteroscedasticity using Kendal rank correlation coefficient τ. When τ > 0.1 a logarithmic transformation was performed on the data before calculating bias and 95% upper and lower limits of agreement.” See line 241.

  • line 380: any idea why hip internal rotation in prone has the worst accuracy both with acc and 2D video? Please comment.

Author response: Thank you very much for this comment. We believe that it relates to the knee angle that is difficult to maintain at 90 degrees during hip rotation in prone, thus when the knee is bent i.e. out of plane, it increases the perspective error. Conversely, during seated position the knee is more rigid and is kept to 90 degrees when the hip is rotating.

Reviewer 2 Report

Dear Authors,

It is an interesting and straight-forward designed project.

Comment:
Information about the app are missing. It would be great if you could add it, e.g. in the appendix.

Question:
- What is the operating system of the App (Android or AppleOS).

Author Response

Reviewer 2

It is an interesting and straight-forward designed project.

Comment:

  • Information about the app are missing. It would be great if you could add it, e.g. in the appendix.

Author response: Thank you for that comment. The app is under development at this stage and is not available for the public to download and install. The app itself is not interactive i.e. the participants just click the button to start accelerometer sampling and to connect to the clinician web browser application. At this stage the web application does not provide any sophisticated analysis and reporting.

Question:

  • What is the operating system of the App (Android or AppleOS).

Author response: Thank you for drawing our attention to this. We included the list the smartphones used by participants by adding the following sentences: “The update frequency of the smartphone raw acceleration was set to maximum, which is usually at least 100 Hz for iPhone (Apple Inc, https://developer.apple.com/documentation/coremotion/getting_raw_accelerometer_events) and android e.g. Samsung and Huawei [12, 13]. Thus, the TelePhysio app was coded to sample at 100Hz. The smartphones used in this study include: iPhone 8, iPhone 8 plus, iPhone 12 Pro Max, iPhone SE 2020, iPhone X, iPhone 11, iPhone 13, Huawei P10 Lite, Huawei P10, Google pixel 6, Samsung Galaxy s20fe, Samsung Galaxy A8, and Xiaomi POCO X3 NFC. At the end of the assessment, both the assessor and the participant can login to the web-based application and review the outcomes. The video recording was performed using the webcam embedded in the participant’s laptop with a resolution of 1920x1080 and 30 Hz sampling rate.” See line 108

Round 2

Reviewer 1 Report

thank you for being amenable to my suggestions. Regards